# Selective Stimulus Intensity during Hotspot Search Ensures Faster and More Accurate Preoperative Motor Mapping with nTMS

**DOI:** 10.3390/brainsci13020285

**Published:** 2023-02-08

**Authors:** Luca Sartori, Samuel Luciano Caliri, Valentina Baro, Roberto Colasanti, Giulia Melinda Furlanis, Alberto D’Amico, Gianluigi De Nardi, Florinda Ferreri, Maurizio Corbetta, Domenico d’Avella, Luca Denaro, Andrea Landi

**Affiliations:** 1Academic Neurosurgery, Department of Neuroscience, University of Padova, 35128 Padova, Italy; 2Department of Neurosurgery, Padova University-Hospital, 35128 Padova, Italy; 3Clinical Neurophysiology, Department of Neuroscience, University of Padova, 35128 Padova, Italy; 4Unit of Neurology, Unit of Clinical Neurophysiology and Study Center of Neurodegeneration (CESNE), Department of Neuroscience, University of Padova, 35128 Padova, Italy; 5Department of Clinical Neurophysiology, Kuopio University Hospital, University of Eastern Finland, 70211 Kuopio, Finland; 6Padova Neuroscience Center (PNC), University of Padova, 35128 Padova, Italy; 7Department of Neuroscience, Venetian Institute of Molecular Medicine (VIMM), University of Padova, 35128 Padova, Italy

**Keywords:** navigated transcranial magnetic stimulation, nTMS, motor mapping, preoperative mapping, hotspot, motor evoked potential, brain stimulation

## Abstract

Introduction: Navigated transcranial magnetic stimulation (nTMS) has emerged as one of the most innovative techniques in neurosurgical practice. However, nTMS motor mapping involves rigorous steps, and the importance of an accurate execution method has not been emphasized enough. In particular, despite strict adherence to procedural protocols, we have observed high variability in map activation according to the choice of stimulation intensity (SI) right from the early stage of hotspot localization. We present a retrospective analysis of motor mappings performed between March 2020 and July 2022, where the SI was only chosen with rigorous care in the most recent ones, under the guide of an expert neurophysiologist. Materials and methods: In order to test the ability to reduce inaccurate responses and time expenditure using selective SI, data were collected from 16 patients who underwent mapping with the random method (group A) and 15 patients who underwent mapping with the proposed method (group B). The parameters considered were resting motor threshold (%), number of stimuli, number of valid motor evoked potentials (MEPs), number of valid MEPs considered true positives (TPs), number of valid MEPs considered false positives (FPs), ratio of true-positive MEPs to total stimuli, ratio of true-positive MEPs to valid MEPs, minimum amplitude, maximum amplitude and mapping time for each patient. Results: The analysis showed statistically significant reductions in total stimulus demand, procedural time and number of false-positive MEPs. Significant increases were observed in the number of true-positive MEPs, the ratio of true-positive MEPs to total stimuli and the ratio of true-positive MEPs to valid MEPs. In the subgroups analyzed, there were similar trends, in particular, an increase in true positives and a decrease in false-positive responses. Conclusions: The precise selection of SI during hotspot search in nTMS motor mapping could provide reliable cortical maps in short time and with low employment of resources. This method seems to ensure that a MEP really represents a functionally eloquent cortical point, making mapping more intuitive even in less experienced centers.

## 1. Introduction

Over the past decade, navigated transcranial magnetic stimulation (nTMS) has emerged as one of the most innovative techniques in daily clinical neurosurgical practice. The large spread of this method relies on its potential to perform the tangible mapping of some neurologic functions before surgery [1,2,3,4,5,6]. Among all, the mapping of motor areas is the most described by authors and has shown an accurate match with direct cortical stimulation (DCS), which remains the gold standard for functional monitoring [3,4,7,8]. This accordance has helped fuel the current role of nTMS motor mapping in neuro-oncological surgery, where the preservation of function is mandatory and a scrupulous preoperative study could reduce the risk of deficits after surgery [9,10,11,12].

Despite its enlarging use in clinical practice, the importance of an accurate method of execution has not been emphasized enough. From a methodological perspective, motor mapping by means of nTMS involves rigorous steps that inevitably produce quantitative parameters. These parameters (e.g., motor hotspot location and resting motor threshold) express high variability and need continuous analysis to be properly understood [13,14,15,16,17]. In particular, despite strict adherence to the procedural protocols cited in the literature, we observed variations in map activation, which sometimes exceeded reasonable anatomical boundaries (Figure 1). We especially noticed that in some patients, the representations of functional motor areas were either too large or too small to be considered reliable. Therefore, we hypothesized that this variability may be due to false-positive motor evoked potentials (MEPs), which, if not intercepted, can lead to errors in evaluation and surgical planning. Their continuous examination is mandatory to avoid misinterpretation, especially in less experienced centers, but involves an excessive lengthening of procedural time for a technique ancillary to surgery.

In our series, we noticed that the choice of stimulation intensity (SI) from the earliest stages of mapping influenced the variability of parameters and results. In particular, a diligent selection of SI during motor hotspot localization, sometimes mentioned in the literature but never emphasized [18,19,20], appears to be a crucial moment to avoid hyper-elicitation phenomena and to reduce false-positive MEPs in later stages. Therefore, as more centers routinely use nTMS, it becomes increasingly important to establish practices to ensure accuracy.

This study presents a retrospective critical review of our series of mappings performed between March 2020 and July 2022, for which the stimulation intensity was chosen with rigorous care only in the most recent ones, in order to test the method’s ability to reduce inaccurate responses and time expenditure.

## 2. Materials and Methods

### 2.1. Methodological Differences

MRI acquisition with a 3T scanner (Ingenia 3T, Philips Healthcare) was performed according to our specific protocol designed for preoperative planning study and described in a previous report [21]. After obtaining 3D T1-weighted images (TR/repetition time = 8, TE/echo time = 3.7), motor mapping was performed with a figure-eight coil stimulator (NBS system 4.3-Nexstim Oy; Elimäenkatu 9 B, Helsinki, Finland). Muscle activity was examined using electromyography surface electrodes and recorded with the nTMS integrated system (integrated eXimia EMG; sampling frequency, 3 kHz/channel; analysis time, 10 ms pre-stimulus and 100 ms post-stimulus; filter bandwidth, 10–250 Hz). The first dorsal interosseus (FDI) for the upper limbs and the tibialis anterior (TA) for the lower limbs were the recorded muscles.

Each mapping started from specific anatomical landmarks on the cortical surface based on Penfield’s description and was performed according to international indications [20]. After April 2021, we changed the modality of hotspot search to test the ability of SI selection to reduce false-positive responses (Figure 2). Indeed, the motor hotspot is the functional core of the eloquent area and represents the place where the resting motor threshold (rMT) is then calculated. The rMT defines the minimum threshold of cortical excitability and guides the stimulation intensity during definitive mapping. Usually, the intensity used during this phase is set with an estimated suprathreshold value generating a cortical electric field of 80–100 V/m; this, however, can also be obtained using different random intensities.

Based on our hypothesis, we performed hotspot search by starting with a standard intensity value (generally, 35%, expressed as the percentage of the maximal stimulation output of the machine), monitoring latencies (i.e., for upper limbs: normal, 15–25 ms; or 10–50 ms, corrected for age and height [14,22,23,24]) and amplitudes of the resulting muscle responses. Then, as in a feedback scheme, we adjusted the SI according to the characteristics of the MEPs as follows:(1)If their amplitude was >500 μV, the stimulation intensity was decreased by 1–2% of the stimulator power up to the range of 100–500 μV;(2)If their amplitude was <100 μV, the stimulation intensity was increased by 1–2% of the stimulator power up to the range of 100–500 μV.

During the procedure, the muscle groups had to be relaxed (as assessed by continuous EMG tracking; the peak-to-peak noise level had to be less than 50 μV), and the coil had to be oriented perpendicularly to the cortex and in the frontal direction. In theory, this expedient allowed us to define the correct hotspot location using the lowest possible SI and avoid subsequent cortical hyper-elicitation phenomena.

### 2.2. Population Characteristics

All patients who underwent nTMS mapping at Department of Neurosurgery of University of Padova since March 2020 were prospectively entered into a database available for subsequent retrospective analysis. Data were collected from 31 patients, of whom the first 16 underwent mapping without regard to stimulation intensity selection (group A) and the second 15 underwent mapping with the proposed method (group B). These patients correspond to 29 and 24 hemispheres investigated, respectively, for a total of 31 and 32 motor areas analyzed. In total, 8 patients out of 31 had a spinal cord pathology, so the mapping of upper limbs (ULs) and lower limbs (LLs) was performed bilaterally. Since the mapping of the orbicularis oris was not performed in spinal cord patients for anatomical reasons, only these two areas were considered in patients with supratentorial lesions for uniformity of analysis. A total of 39 areas of the upper limbs and 24 of the lower limbs were evaluated (Table 1).

### 2.3. Mapping Characteristics

The parameters considered for each map were rMT (%), number of stimuli, number of valid MEPs (amplitude > 50 μV), number of valid MEPs considered true positives (TPs), number of valid MEPs considered false positives (FPs), ratio of true-positive MEPs to total stimuli, ratio of true-positive MEPs to valid MEPs, minimum amplitude, maximum amplitude and mapping time for each patient. According to the international protocol [20] and case series described in the literature, definitive mapping was conducted in all patients using a stimulation intensity of 110% rMT and a 5 × 5 mm grid by stimulating above the intersections. Each individual mapping area was considered finished when a double grid line of negative spots was detected on its outer edge, corresponding to an absent response or to MEPs of excessively low amplitude (<50 μV).

### 2.4. MEP Discrimination Process

False-positive MEPs are non-physiological motor responses of amplitude higher than >50 μV but with incorrect latency or shape, mainly due to artifactual or hyper-elicitation phenomena. However, they are not excluded by the nTMS machine, which instead considers them functional signals. The purpose of this analysis was to verify that only cortical locations that elicited true-positive MEPs were accepted in the map. Therefore, each MEP was critically examined by three team members, an experienced neurophysiology technician (G.D.N.), a senior (A.L.) and a younger neurosurgeon (L.S.), to discriminate true- from false-positive responses. Discrimination took place in two separate stages: In the first stage, the three professionals performed the assessment anonymously and independently. In the second stage, they reevaluated all the maps together to confirm their choices. The main elements that determined whether the responses were real potentials or artifacts were latencies, amplitudes and their wave forms (Figure 3).

### 2.5. Statistical Analysis

The two groups were compared to detect any statistical difference in baseline factors as well as in the accuracy and efficiency of motor mapping. In addition, subgroup analyses were carried out only considering upper-limb maps, lower-limb maps, patients with brain tumors or those with medullary lesions. Statistical analyses were performed using SPSS software (version 20; SPSS Inc., Chicago, IL, USA). The Pearson chi-square test was used for discrete variables, and the t-test, for continuous ones. The statistical significance was set at *p* < 0.05.

### 2.6. Patient Informed Consent and Ethical Approval

The patients signed their specific informed consent for MRI acquisition and nTMS examination. The study was conducted in accordance with the ethical standards of Institutional Research Committee AOUP (Prot. No. 0024711 (07/04/2022) and Prot. No. 0043481 (27/06/2022)) and with the 1964 Declaration of Helsinki, plus later amendments.

## 3. Results

From 1 March 2020 to 31 July 2022, 31 consecutive patients underwent nTMS motor mapping performed by two experienced authors (L.S. and S.L.C.). In total, 23 patients out of 31 (74.2%) had a brain lesion in the eloquent area (20 telencephalic lesions and 3 diencephalic lesions), and mapping was performed in order to plan the surgical strategy. The other eight patients (25.8%) suffered from a medullary lesion with clinical myelopathy (three intrinsic lesions and five degenerative cervical myelopathy cases), and nTMS was performed with a research interest according to recent perspectives in the literature [25]. Excluding cases with degenerative cervical pathology, 17 patients (65.4%) had gliomas (8 low-grade gliomas and 9 high-grade gliomas (WHO 2021)); a total of 3 patients (11.5%) had metastatic tumors; a total of 2 patients (7.7%) had meningiomas; and 4 patients (15.4%) had other conditions (2 medullary ependymomas, 1 medullary hemangioblastoma and 1 parietal cavernoma).

At examination time, the mean age was 46.3 years (range of 11–79 years). There were 14 female patients (45.2%) and 17 male patients (54.8%).

nTMS was performed in both hemispheres in the 8 patients with medullary lesions and only in the pathological one in the other 23 patients with encephalic neoplasms. Overall, we collected 63 motor areas, 39 (61.9%) for upper limbs and 24 (38.1%) for lower limbs. Comparing the maps derived using the two methods, we did not observe significative differences in terms of group characteristics (gender, *p* = 0.921; hemisphere location, *p* = 0.379; limb relation, *p* = 0.674; lesion site, *p* = 0.153). In particular, 61.3% of cases in group A and 62.5% in group B were male; in contrast, female patients were 38.7% of group A and 37.5% of group B. Regarding hemisphere analysis, we performed mapping in the right one in 54.8% of group A and 43.8% of group B. The left hemisphere was analyzed in 45.2% of patients in group A and 56.2% in group B. Area-specific mapping was performed in 64.5% for the upper limbs and 35.5% for the lower limbs in group A, and 59.4% and 40.6% in group B, respectively. Lastly, the lesion was localized into the brain in 67.7% of cases in group A and intramedullary in the remaining 32.3%, while in group B, it was equally divided into 50% and 50% (Table 2).

In both groups, we noted missing maps that we did not find despite careful modulation of coil orientation and position above brain sulci, since the coil tilt or orientation can affect the mapping results [26,27,28]. In all the cases, we only experienced difficulties during lower-limb mapping. Regarding maps from group A, the elicitation of MEPs from TA muscle was unsatisfactory in two patients with glioma, bilaterally in two patients with degenerative cervical myelopathy, and in just one hemisphere in one other patient with degenerative cervical myelopathy and in one patient with Rolandic meningioma. In contrast, in group B, we did not obtain responses in four patients with glioma strictly adjacent to this motor area and in one Rolandic meningioma. In one patient in group B, we did not search the lower-limb area due to its high distance from the glioma location.

Comparing the rMT values in all maps, no differences were observed in the two groups, with an average value of 47.90% in group A and 47.16% in group B (*p* = 0.846).

### 3.1. Full-Sample Analysis

The univariate analysis showed, in group B, statistically significant reductions in total stimulus demand, procedural time and number of false-positive MEPs (Table 3). The comparison revealed that the number of stimuli required with the SI selection method was significantly smaller than that required with the random stimulation strategy (*p* < 0.0005), with an average of 67.97 stimuli versus 110.16 stimuli. Additionally, the number of false-positive responses differed consistently, with an average of 8.31 MEPs in group B and 34.35 MEPs in group A (*p* < 0.0005). Furthermore, although not expected, the choice of SI during hotspot search exhibited a statistically significant effect on the absolute procedural time for mapping, with an average of 62.93 min versus 104.50 min (*p* < 0.0005).

It is worth noting that significant differences were observed when comparing the number of true-positive MEPs (*p* < 0.050), the ratio of true-positive MEPs to total stimuli (*p* < 0.0005) and the ratio of true-positive MEPs to valid MEPs (*p* < 0.0005). In particular, the true-positive responses for each area were increased in group B, with an average of 32.09, compared to 25.26 in group A. In contrast, the ratio of true-positive MEPs to total stimuli, which represents the diagnostic sensibility of the technique, was significantly higher in group B, where we found an average value of 47.18% versus 24.19% in group A. In addition, when comparing the ratio of true-positive MEPs to all valid MEPs, which describes the predictive positive value of the method, a significant increase was observed in group B, with an average probability of 79.27% that a motor response related to a real eloquent cortical spot, versus a value of 43.73% in group A.

Furthermore, analyzing a possible effect of SI choice on MEP electrical parameters, we compared their extreme amplitude values (min–max), and as expected, we did not find differences (min, *p* = 0.832; max, *p* = 0.858) (Figure 4).

### 3.2. Subgroup Analysis

Subsequently, we divided all maps into four smaller cohorts according to motor function (i.e., upper limb and lower limb) and lesion site (i.e., brain lesion and medullary lesion) and performed a comparison. Their univariate analyses showed similar trends to the one performed on the whole sample, and in particular, the increase in true positives and decrease in false positives were confirmed in all subgroups analyzed (Figure 5).

#### 3.2.1. Upper-Limb Sample

Total stimulus demand showed a reduction in group B, 77.05, compared with group A, 128.15, (*p* = 0.001), as did the number of false-positive MEPs, which was 9.32, compared with 36.90 (*p* < 0.0005). Although at the limits of statistical significance, an increase in the number of true-positive MEPs was observed from 30.50 in group A to 38.84 in group B (*p* = 0.051). In contrast, we found that the average ratio of true-positive MEPs to total stimuli and the average ratio of true-positive MEPs to all valid MEPs maintained significant increases from 26.51% to 50.91% and from 47.80% to 81.25%, respectively (*p* < 0.0005).

#### 3.2.2. Lower-Limb Sample

The analysis confirmed that there were no differences in average rMT (61.82% in group A and 62.62% in group B, *p* = 0.875) and extreme amplitude values (min, *p* = 0.567; max, *p* = 0.763). Similar to the upper-limb case, the stimulus demand and the number of false-positive MEPs showed reductions in group B compared with group A (54.69 versus 77.45 (*p* = 0.009) and 6.85 versus 29 (*p* < 0.0005)). An increase bordering statistical significance in the number of true-positive MEPs was observed in group B, where the average value was 22.23, compared with 15.73 in group A (*p* = 0.055), while stronger differences were found in the ratio of true-positive MEPs to total stimuli and the ratio of true-positive MEPs to all valid MEPs, from 19.98% to 41.73% and from 36.35% to 76.39%, respectively (*p* < 0.0005).

#### 3.2.3. Brain-Lesion Sample

In group B, total stimulus demand and the number of false-positive MEPs showed significant reductions, with average values of 72.44 compared with 107.90 (*p* = 0.015) and of 10.38 compared with 37.57 (*p* < 0.0005). On the contrary, the increases in the number of true-positive MEPs (*p* = 0.022), the ratio of true-positive MEPs to total stimuli (*p* < 0.0005) and the ratio of true-positive MEPs to all valid MEPs (*p* < 0.0005) were confirmed to be from 23.86, 23.37% and 39.82% in group A to 33.19, 48.12% and 75.75% in group B, respectively.

#### 3.2.4. Medullary-Lesion Sample

Stimulus demand and the number of false-positive MEPs showed reductions in group B compared with group A (63.50 versus 114.90 (*p* = 0.003) and 6.25 versus 27.6 (*p* < 0.0005)). An increase in the number of true-positive MEPs was observed in group B, although it was not statistically significant, with an average value of 31, compared with 28.2 in group A (*p* = 0.671). More significant increases were found in the ratio of true-positive MEPs to total stimuli, from 25.92% to 46.24% (*p* = 0.002), and in the ratio of true-positive MEPs to all valid MEPs, from 51.96% to 82.80% (*p* < 0.0005).

## 4. Discussion

The spreading of nTMS motor mapping in a growing number of neurosurgical units confirms its clinical value for the surgical management of brain tumors in the eloquent area [29]. The possibility to have objective spatial information before surgery about the relationship between lesion and functional tissue makes this tool increasingly attractive. In particular, as shown in the recent literature, its greatest strength is the ability to improve motor and oncologic outcomes [30,31].

However, motor mapping is a multi-step technical examination, and the wrong interpretation of its quantitative parameters can undermine its reliability, leading to inaccurate results that can cause harmful consequences for the patient. Previous studies showed a tendency to standardize the process itself, revealing the need for a definitive protocol to make it easily repeatable in all centers. These papers focused on a high variability of parameters and tried to normalize them with different technical expedients, such as the correction of stimulation techniques or the application of different amplitude criteria for response discrimination [32,33,34]. However, we noticed that this effort was always limited to the last stage of mapping, without considering the possible influence of the initial stage. The initial mapping is, however, a very important part of the examination, because it defines the hotspot location and, consequently, the rMT value. To our knowledge, this paper is the first that shifts the focus to the first steps in order to achieve even more accurate motor mapping.

For illustrative purposes, the mapping process can be divided into two stages: (1) preparation for mapping and (2) definitive mapping. During the first stage, the operator performs a rough stimulation of the motor cortex to discover the functional core of that specific area, precisely named hotspot, where they later calculate the rMT [35]. This parameter defines the minimum threshold of cerebral cortical excitability that subsequently guides the intensity of the second mapping phase. The literature shows that intensities slightly above the resting motor threshold (105–120% of rMT) have the advantage of stimulating sufficient cortical volume, obtaining more precise functional maps [36]. This happens because the selection of higher intensities used in definitive mapping enlarges the map, increasing the spread of responses and losing the spatial accuracy of individual stimuli [36,37,38]. Not surprisingly, at the beginning of the millennium, Di Lazzaro et al. already showed that the amplitude of MEPs and the amount of excited motor neurons depend on the intensity of stimulation [39].

From this perspective, hotspot location and rMT value are probably the cornerstones of the whole process, thus needing precise guidance. We empirically observed their variability, especially when we used high stimulation intensities in the pre-mapping phase, apparently making the final result less reliable.

Regarding hotspot location, several studies showed that it can be various and that it can be found away from its expected position on the cortical surface [40,41,42,43,44,45]. In addition, in a comparative study regarding hotspot location of the hand motor area in healthy subjects and patients with chronic neuropathic pain, Ahdab et al. showed that in some patients, the MEP amplitude could be higher in premotor than primary motor cortex (M1) stimulation; thus, the hotspot can be erroneously found beyond the anatomical edge of M1, mistaking the premotor cortex for the motor cortex [46]. However, similar findings were reported years ago by other authors, showing that different areas of the primary motor cortex may be involved in the induction of motor function [47] and that the non-primary motor cortex may significantly contribute to MEP generation, leading to great variation in hotspot location [28,48]. In addition, it was confirmed that nTMS is able to produce MEPs with higher probability than the non-navigated one [49,50] but also to increase the probability of obtaining potential even beyond the precise target. Thus, it is essential to determine the accurate target of nTMS, because MEP elicitation itself may or may not correspond to a correct eloquent point.

The main goal of preoperative mapping is to identify cortical points that, if injured during surgery, would cause an irreversible motor deficit. These points belong to the primary motor cortex, where it is essential to avoid gross localization errors that could lead to the misinterpretation and underestimation of surgical risk. However, the nTMS machine does not indicate whether the obtained MEP really belongs to the primary motor cortex, and if not, the mapped area loses reliability.

Furthermore, it has been shown that a near-threshold intensity could reduce the variability of individual motor map location during nTMS procedures [36,51], because lower SI could allow one to more selectively recruit the cluster of pyramidal neurons at the center of the primary motor cortex and corticospinal tract. Moreover, in addition to this widespread cortical activation, the influence of a second long-lasting effect is likely. Nevertheless, in our experience, starting with high SI led to excessive easiness in evoking MEPs during definitive mapping, and often not only within M1, as if cortical excitability had increased. Not surprisingly, the literature has shown a similar effect with other neurophysiological methods where it was well described, such as in transcranial direct cortical stimulation (tDCS) [52,53,54], but for now, a similar mechanism in nTMS can only be hypothesized.

Therefore, taking advantage of this information, we reduced our SI during hotspot search, assuming a lower influence of non-primary motor cortex activation. A limited number of studies have shown that motor maps obtained with high stimulation intensity cause hyper-elicitation in motor areas, especially where rMT is usually low, as in the case of the distal muscles of the hand or foot [32,55,56]. This happens because the cortex is affected by a wide distribution of the electric field, which stimulates nearby areas by expanding estimates of representations [57]. The effect leads to a variation in the size of the area with the invasion of nearby non-primary motor areas and the inability to detect its plastic changes. van de Ruit et al. and Thordstein et al. reported differences in the effect of stimulation intensity on the map area, in particular those of the first dorsal interosseus (FDI), the abductor pollicis brevis (APB) and the tibialis anterior (TA) muscles, showing that the SI should be carefully chosen based on the aim of the mapping procedure [32,37]. However, no work has paid attention to the SI during the initial phase of motor mapping. In fact, if the SI affects the spread of the current in the motor cortex [55], it could also have an effect during hotspot search.

Based on the results obtained using the SI method, we found a significant reduction in the number of false-positive responses, demonstrating a likely influence of intensity not only on the last steps (after rMT calculation) but also on the entire process.

In addition, considering the ratios between the quantitative values of our results, we demonstrated notable increases in the sensibility and the predictive positive value of the method, expressed by the ratio of true-positive MEPs to total stimuli and the ratio of true-positive MEPs to all valid MEPs, respectively. From a purely practical point of view, these results could be an invaluable aid for physicians beginning to approach this preoperative mapping tool. Indeed, a sensibility value of 47.18% means that about one out of two stimuli corresponds to a spot of true cortical eloquence, and a predictive positive value of 79.27% means that a valid MEP has an extremely high probability of being a true-positive MEP. In this way, the result is more reliable and, especially, does not need more time to be re-examined. Unexpectedly, total stimulus demand and the time needed for the whole mapping were decreased in a significant way, leading the nTMS procedure to be more comfortable for the patients and for the operator. Therefore, in our opinion, the real advantage brought by the SI selection method is the greater intuitiveness that the combination of these small results provides, reducing revision work and producing maps with marked distinction between eloquent and non-eloquent areas.

Furthermore, we observed a uniform reduction in the number of stimuli required for a single mapping, which might express an indirect reduction in area size. As a matter of fact, the lack of uniformity in the area of the maps during our first attempts was a crucial element that led to the decision to change the mapping method. Using a grid of 5 × 5 mm and only giving pulses over the line intersections up to two negative lines, we obtained maps with more defined edges that, for this reason, must be smaller when composed with fewer stimuli. The redundant false-positive stimuli, which showed a significant reduction in group B, probably did not belong to the primary motor area but represented non-eloquent spots that, in group A, were mainly located at the edges and periphery. The size of the area has been previously described and accurately compared, but the proposed method allows a quick and easy evaluation to be performed and facilitates the reduction in the cortical spread of the stimulus during mapping.

In clinical practice, the neurosurgeon is particularly interested in reliable knowledge of the boundaries surrounding the motor area to best plan the approach. Although other works, more complex and intended primarily for technicians, have presented methods that greatly improve mapping [15,57,58,59,60,61,62], a universally usable variant able to guarantee a useful result for surgical purposes has never been described.

Nevertheless, nTMS motor mapping in neurosurgery no longer only has value in surgical planning. The first added value was its capacity to provide preoperative risk stratification thanks to the careful evaluation of mapping and tractography [63]. The risk stratification model, initially proposed by the Berlin group and later confirmed by other centers [64,65,66], divides lesions into low and high surgical risk according to their motor cortex infiltration, the distance from the corticospinal tract (≤8 mm or >8mm) and the ratio of the rMT of both hemispheres (<90%/>110%). Therefore, given the importance of the distance between lesions and functional areas, proper risk stratification depends on the reliable reproduction of the motor area and, in particular, its boundaries. In contrast, the second implementation was the recent application of the interpretation of intraoperative monitoring (IOM) phenomena. Confounder phenomena are indeed common during IOM and could influence surgical choice. In fact, reversible alterations in MEP amplitude or irreversible decreases ≤ 50% during surgery are frequent, warning elements that cause the suspension or discontinuation of resection. The incorrect assessment of these phenomena can impact both motor outcomes and the extent of surgical resection. Recently, Rosenstock et al. proposed a prognostic correlation between preoperative nTMS risk stratification [64] and IOM alterations in the amplitude of MEPs, which provides support for the interpretation of ambiguous phenomena and the regulation of DCS [12]. Therefore, preoperative knowledge of the true extent of the functional areas allows one to make clearer intraoperative decisions without limiting tumor excision and especially by estimating reduced postoperative risk of irreversible motor deficit.

However, although this study could be a convenient aid for clinicians in daily preoperative activities, it presents some limitations. The first limitations are the retrospective design, which may hide biases in patient selection, and the small sample size. Another limitation is that the two SI selection methods have never been evaluated in the same patient. Future research should be conducted to compare the influence of the SI selection method on cortical excitability in the same patient in order to increase the knowledge about the influence of interindividual factors, which was not fully considered in our results. Further, although the mappings were not performed by a single operator, the study did not consider the potential influence of the learning curve, which can unintentionally optimize the entire procedure, especially in terms of time spent and decision making.

Although the proposed method resulted in an improvement of our results during mapping, it cannot yet explain the complex mechanism underlying the motor system and its cortical representation on its own. So far, neuroplasticity and cortical excitability mechanisms hide other attractive secrets and questions that our study only stimulated. For example, we encountered difficulties during lower-limb mapping, as in other series [8,67,68,69], both with the traditional method and the new method. Excluding two patients with meningioma, in whom the size of the lesion did not make transcranial stimulation feasible, the mapping of the tibialis anterior muscle in the other cases did not improve when using a higher SI, which was nevertheless elevated to the maximum power of the nTMS stimulator. Six patients with glioma without motor deficit were operated, and in all cases, positive spots were only observed during intraoperative stimulation in the lower part of the surgical site, confirming the negativity of the cerebral cortex. These findings, although observational, confirm that there are several factors besides SI that influence the feasibility of mapping with nTMS, and these should be the subject of further investigation, particularly for the lower extremities.

## 5. Conclusions

The precise selection of stimulation intensity during hotspot search in motor mapping with nTMS could be a viable option to obtain reliable cortical maps with low employment of time and resources. This method seems to ensure with remarkable effectiveness that a motor evoked potential really represents a functionally eloquent cortical point, making mapping more intuitive even in less experienced centers. Moreover, an accurate prediction of the surgical risk based on reliable motor mapping could allow even safer and complete resection to be performed and provide an accurate perspective on surgical risk to the patient. This work could be an incentive for the development of subsequent studies to validate our results in a more homogeneous population or by testing the two methods on the same subjects.

## Figures and Tables

**Figure 1 brainsci-13-00285-f001:**
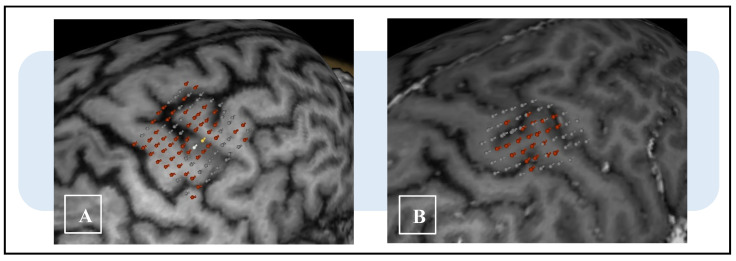
Example of variability of cortical mapping area of the first dorsal interosseus (FDI) in two different patients with eloquent subcortical high-grade gliomas. The different distribution and the extent of cortical positivity points are evident. (**A**) The map shows positive responses beyond the reasonable boundaries of activation, hiding the limit of the eloquent area. (**B**) The points of activation are demarcated by a surrounding, non-eloquent area, later confirmed during surgery. The differently colored points indicate the amplitude of MEPs (red, 50–500 μV; yellow, 500–1000 μV; white, ≥1 mV; gray, ≤50 μV or no response).

**Figure 2 brainsci-13-00285-f002:**
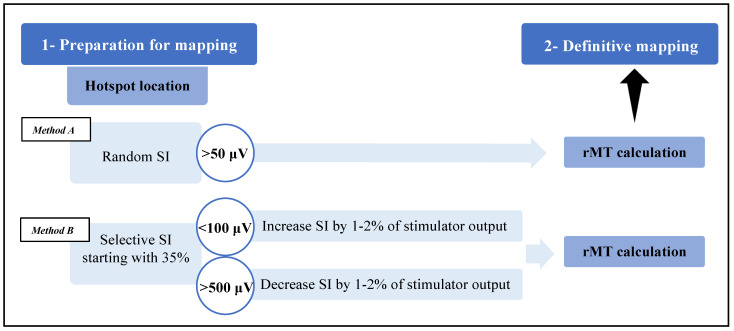
Schematic representation of mapping phases with two different hotspot search methods. The procedural scheme ranges from an initial preparation phase to final mapping via hotspot search and rMT calculation. The hotspot search method in group B considers MEP amplitudes to adjust stimulation intensity (SI), which starts from a standard value of 35%, while in group A, SI is chosen randomly and does not consider response characteristics.

**Figure 3 brainsci-13-00285-f003:**
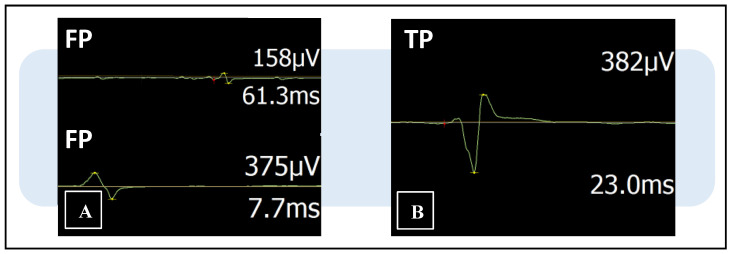
Examples of three motor evoked potentials (MEPs) displayed in the nTMS integrated EMG system with trace line, amplitude and latency values. The machine cannot recognize false-positive responses, which are instead erroneously considered eloquent points. (**A**) False-positive MEPs (FP) of correct amplitude (>50 μV) but incorrect latency and shape. (**B**) True-positive MEP (TP) with regular shape, amplitude and latency value.

**Figure 4 brainsci-13-00285-f004:**
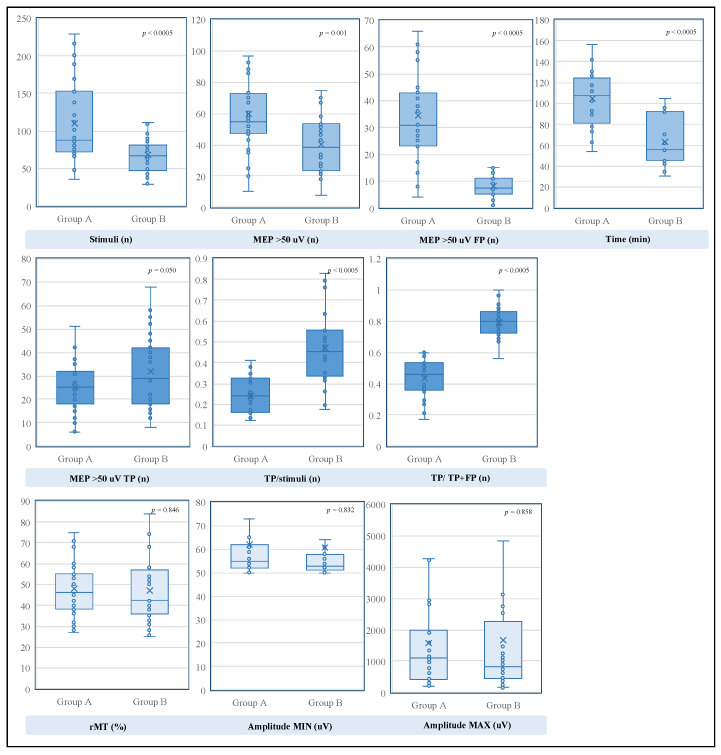
Graphical representation of the comparison between the parameters of group A and group B with their respective significance values. At the top, the parameters that showed significant decreases are grouped. In the middle, the parameters that increased are shown. In the lower part, the parameters that did not show significant differences are grouped. The statistical significance was set at *p* < 0.05 (MEP, motor evoked potential; TP, true positive; FP, false positive; rMT, resting motor threshold).

**Figure 5 brainsci-13-00285-f005:**
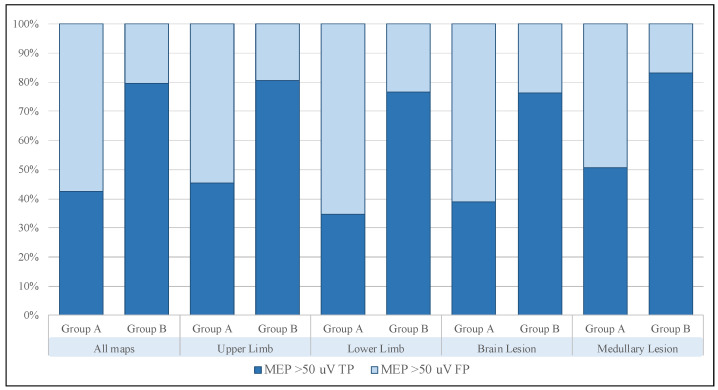
Schematic representation of true and false responses in the full sample and all subgroups. The trend is similar in all cases, and the analysis shows a steady increase in the ratio of true- to false-positive MEPs. (MEP, motor evoked potential; TP, true positive; FP, false positive).

**Table 1 brainsci-13-00285-t001:** Overview of the whole patient sample and pathologies.

Population Characteristic	Total (*n* = 31)
Mean age, years (min–max)	46.30 (11–79)
Gender, *n* (%) Female Male	14 (45.2) 17 (54.8)
Pathology, *n* (%) Brain Telencephalic Diencephalic Medullary Intrinsic lesion Degenerative cervical myelopathy	23 (74.2) 20 (86.9) 3 (13.1) 8 (25.8) 3 (37.5) 5 (62.5)
Oncological grading, *n* (%) Glioma I–II III–IV Metastases Meningioma Others	17 (65.4) 8 (47.1) 9 (52.9) 3 (11.5) 2 (7.7) 4 (15.4)

**Table 2 brainsci-13-00285-t002:** Distribution of maps in the two groups according to patient characteristics, brain area and lesion location.

Map Characteristic	Total (*n* = 63)	Group A (*n* = 31)	Group B (*n* = 32)	*p* Value
Mean age, years (min–max)	46.30 (11–79)	48.10	44.56	-
Gender, *n* (%) Female Male	24 (38.1) 39 (61.9)	12 (38.7) 19 (61.3)	12 (37.5) 20 (62.5)	0.921
Hemisphere, *n* (%) Right Left	31 (49.2) 32 (50.8)	17 (54.8) 14 (45.2)	14 (43.8) 18 (56.3)	0.379
Limb, *n* (%) Upper Lower	39 (61.9) 24 (38.1)	20 (64.5) 11 (35.5)	19 (59.4) 13 (40.6)	0.674
Lesion site, *n* (%) Brain Medullary	37 (58.7) 26 (41.3)	21 (67.7) 10 (32.3)	16 (50) 16 (50)	0.153

**Table 3 brainsci-13-00285-t003:** Results of mapping parameters in the two groups and statistical analysis; the upper part shows the results of the maps of the full sample, while the lower part shows those of the subgroups. The statistical significance was set at *p* < 0.05.

**Mapping Characteristic.** **Full Sample**	**Total** **(*n* = 63)**	**Group A** **(*n* = 31)**	**Group B** **(*n* = 32)**	***p* Value**
rMT, value (min–max)	47.52% (25–92%)	47.90%	47.16%	0.846
Stimuli, n (min–max)	88.73 (29–229)	110.16	67.97	<0.0005
MEPs >50 uV (min–max) >50 uV TP (min–max) >50 uV FP (min–max)	49.86 (8–119) 28.73 (6–68) 21.13 (0–87)	59.61 25.26 34.35	40.41 32.09 8.31	0.001 0.050 <0.0005
TP/stimuli, % (min–max)	35.87 (12.5–93.33)	24.19	47.18	<0.0005
TP/TP+FP, % (min–max)	61.79 (17.14–100)	43.73	79.27	<0.0005
Amplitude, uV (min–max) min max	61.27 (50–176) 1626.81 (163–8465)	61.84 1585.55	60.72 1666.78	0.832 0.858
Time, min (min–max)	84.39 (30–156)	104.50	62.93	<0.0005
**Mapping characteristic** **Upper Limb**	**Total** **(*n* = 39)**	**Group A** **(*n* = 20)**	**Group B** **(*n* = 19)**	***p* value**
rMT, value (min–max)	38.46% (25–55%)	40.25%	36.58%	0.148
Stimuli, n (min–max)	103.26 (30–229)	128.15	77.05	0.001
MEPs >50 uV (min–max) >50 uV TP (min–max) >50 uV FP (min–max)	58.03 (8–119) 34.56 (8–68) 23.46 (0–87)	67.40 30.50 36.90	48.16 38.84 9.32	0.010 0.051 <0.0005
TP/stimuli, % (min–max)	38.39 (13.97–93.33)	26.51	50.91	<0.0005
TP/TP+FP, % (min–max)	64.1 (26.89–100)	47.80	81.25	<0.0005
Amplitude, uV (min–max) min max	60.33 (50–176) 2042.13 (163–8465)	59.95 1964.70	60.74 2123.63	0.912 0.815
**Mapping characteristic** **Lower Limb**	**Total** **(*n* = 24)**	**Group A** **(*n* = 11)**	**Group B** **(*n* = 13)**	** *p* ** **value**
rMT, value (min–max)	62.25% (42–92%)	61.82%	62.62%	0.875
Stimuli, n (min–max)	65.13 (29–113)	77.45	54.69	0.009
MEPs >50 uV (min–max) >50 uV TP (min–max) >50 uV FP (min–max)	36.58 (10–70) 19.25 (6–38) 17.33 (3–43)	45.45 15.73 29.73	29.08 22.23 6.85	0.010 0.055 <0.0005
TP/stimuli, % (min–max)	31.76 (12.5–79.31)	19.98	41.73	<0.0005
TP/TP+FP, % (min–max)	58.04 (17.14–88.46)	36.35	76.39	<0.0005
Amplitude, uV (min–max) min max	62.79 (50–119) 951.92 (163–2955)	65.27 896.18	60.69 999.08	0.567 0.763
**Mapping characteristic** **Brain lesion**	**Total** **(*n* = 37)**	**Group A** **(*n* = 21)**	**Group B** **(*n* = 16)**	***p* value**
rMT, value (min–max)	44.76% (25–92%)	46.43%	42.56%	0.454
Stimuli, n (min–max)	92.57 (29–229)	107.90	72.44	0.015
MEPs >50 uV (min–max) >50 uV TP (min–max) >50 uV FP (min–max)	53.70 (21–119) 27.89 (6–58) 25.81 (1–87)	61.43 23.86 37.57	43.56 33.19 10.38	0.017 0.022 <0.0005
TP/stimuli, % (min–max)	354.07 (12.5–93.33)	23.37	48.12	<0.0005
TP/TP+FP, % (min–max)	55.36 (17.14–96.55)	39.82	75.75	<0.0005
Amplitude, uV (min–max) min max	62.03 (50–176) 1818.81 (192–6639)	62.10 1827.81	61.94 1807.00	0.985 0.972
**Mapping characteristic** **Medullary lesion**	**Total** **(*n* = 26)**	**Group A** **(*n* = 10)**	**Group B** **(*n* = 16)**	***p* value**
rMT, value (min–max)	51.46% (28–84%)	51%	51.75%	0.898
Stimuli, n (min–max)	83.27 (35–201)	114.90	63.50	0.003
MEPs >50 uV (min–max) >50 uV TP (min–max) >50 uV FP (min–max)	44.38 (8–94) 29.92 (6–68) 14.46 (0–45)	55.8 28.2 27.6	37.25 31 6.25	0.053 0.671 <0.0005
TP/stimuli, % (min–max)	35.87 (12.5–93.33)	25.92	46.24	0.002
TP/TP+FP, % (min–max)	61.79 (17.14–100)	51.96	82.80	<0.0005
Amplitude, uV (min–max) min max	61.27 (50–176) 1626.81 (163–8465)	61.3 1076.8	59.5 1526.56	0.772 0.547

## Data Availability

All data are available in the text.

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
