# Peer review of "Selective Stimulus Intensity during Hotspot Search Ensures Faster and More Accurate Preoperative Motor Mapping with nTMS"

_brainsci, 2023, doi:10.3390/brainsci13020285_

Round 1

Reviewer 1 Report

1.     Please add the full term in the abstract for acronyms seen at the first time so that readers can understand their meaning instead of searching from main texts

2.     I’m not if it is an online printing issue as I see comas in float numbers. Please replace comma with decimal point in order to fulfill scientific soundness.

3.     In Table 3, why you eliminate Time for subgroup analysis? As time is an important factor in clinical practice

4.     The majority of subjects used in your study have brain lesions. Hence, your conclusion should add this limitation for others to replicate your results
Reviewer 2 Report

A revision is not required

Author Response

Thank you for taking the time to review our manuscript. We are very glad of your positive comment.

Reviewer 3 Report

The manuscript titled ‘Selective stimulus intensity during hotspot search in pre-operative motor mapping with nTMS increases neurosurgical reliability’ by Sartori et al. compared two methods of pre-operative motor mapping through TMS and found selective stimulus intensity method is more accurate and time efficient than random stimulus intensity method.

This important piece of information may be reported briefly compared to its current form. The authors should recheck and eliminate every possibility of repetitions in the figures, tables, and text (especially in the result section).

Title of the study mentions that the method ‘increase neurosurgical reliability’, but it seems that the study has not objectively assessed the reliability (reproducibility) of the surgery. Hence, the authors may have to modify the title.   

The neuro-navigation technology and software used for the study may be briefly described, preferably through an illustration.

The figure 1 should include the timelines of data collection for group A and group B. In fact, it seems Figure 1 has been prematurely introduced in the text before describing two separate methods. Interestingly, in 2020-2021 the traditional method for finding hotspot was used and in 2021-2022 the improved and more careful method was used. Which raises a doubt if the improvement noted in the later part of the assessment was effect of the technique or a result of improved skill of the technicians due to their experience. The authors may address this shortcoming in the manuscript.

The authors are requested to put reference to ‘international indication’ in line 96. Is it an indication or guideline?

How was the cortical electrical field estimated in the study? Or is it taken from previously published literature? For threshold identification, classically the EMG voltage is considered as the cornerstone in non-surgical TMS field.

Ref “Resting-motor threshold (RMT) was assessed using the Magstim Rapid stimulator as the lowest intensity able to evoke a MEP of more than 50 μV in at least five out of ten consecutive trials in the right FDI (Rothwell et al. 1999)” - Francesca Gilio, 2003.

The figure 2 should illustrate the steps of threshold identification in random stimuli method.

I am not entirely sure if the authors have interchangeably used ‘hotspot identification’ and ‘motor threshold identification’ in line 110. If so, it is not correct. Please recheck and confirm.

Line 122 mentions that ‘the coil is placed perpendicular to the cortex’. Please confirm if this is parallel or perpendicular.  

Line 130-131 is not clear to me. Why were the number of hemispheres and number of motor areas not identical?

Please ensure that table 3, figure 4 and figure 5 are entirely exclusive with no repetitions.

Line 443 – the sentence may be paraphrased.

Overall, the study would have been more robust if the authors could get the outcome data related to motor deficit following the surgery. 
